# Studies to Improve Perinatal Health through Diet and Lifestyle among South Asian Women Living in Canada: A Brief History and Future Research Directions

**DOI:** 10.3390/nu13092932

**Published:** 2021-08-24

**Authors:** Dipika Desai, Sujane Kandasamy, Jayneel Limbachia, Michael A. Zulyniak, Paul Ritvo, Diana Sherifali, Gita Wahi, Sonia S. Anand, Russell J. de Souza

**Affiliations:** 1Population Health Research Institute, Hamilton Health Sciences Corporation, Hamilton, ON L8L 2X2, Canada; Dipika.Desai@phri.ca (D.D.); dsherif@mcmaster.ca (D.S.); anands@mcmaster.ca (S.S.A.); 2Department of Health Research Methods, Evidence, and Impact, McMaster University, Hamilton, ON L8S 4L8, Canada; kandas3@mcmaster.ca (S.K.); wahig@mcmaster.ca (G.W.); 3Department of Medicine, McMaster University, Hamilton, ON L8S 4L8, Canada; Jayneel.Limbachia@phri.ca; 4School of Food Science and Nutrition, University of Leeds, Leeds LS2 9JT, UK; M.A.Zulyniak@leeds.ac.uk; 5Department of Psychology, York University, North York, ON M3J 1P3, Canada; pritvo@yorku.ca; 6School of Nursing, McMaster University, Hamilton, ON L8S 4K1, Canada; 7Department of Pediatrics, McMaster University, Hamilton, ON L8S 4L8, Canada

**Keywords:** perinatal, South Asians, nutrition, infant and child health, maternal health, healthy active living, gestational diabetes, culturally-tailored advice, prospective cohort, randomized control trial

## Abstract

South Asians (i.e., people who originate from India, Pakistan, Sri Lanka, Nepal, and Bangladesh) have higher cardiovascular disease rates than other populations, and these differences persist in their offspring. Nutrition is a critical lifestyle-related factor that influences fetal development, and infant and child health in early life. In high-income countries such as Canada, nutrition-related health risks arise primarily from overnutrition, most strikingly for obesity and associated non-communicable diseases. Evidence for developmental programming during fetal life underscores the critical influence of maternal diet on fetal growth and development, backed by several birth cohort studies including the Pune Maternal Nutrition Study, the South Asian Birth Cohort Study, and the Born in Bradford Study. Gestational diabetes mellitus is a strong risk factor for type 2 diabetes, future atherosclerosis and cardiovascular disease in the mother and increases the risk of type 2 diabetes in her offspring. Non-pharmacological trials to prevent gestational diabetes are few, often not randomized, and are heterogeneous with respect to design, and outcomes have not converged upon a single optimal prevention strategy. The aim of this review is to provide an understanding of the current knowledge around perinatal nutrition and gestational diabetes among the high-risk South Asian population as well as summarize our research activities investigating the role of culturally-tailored nutrition advice to South Asian women living in high-income settings such as Canada. In this paper, we describe these qualitative and quantitative studies, both completed and underway. We conclude with a description of the design of a randomized trial of a culturally tailored personalized nutrition intervention to reduce gestational glycaemia in South Asian women living in Canada and its implications.

## 1. Introduction

As of 2016 in Canada, >1.9 million people (5.4% of the population) were of South Asian (SA) ancestry—i.e., people who originate from India, Pakistan, Sri Lanka, Nepal, and Bangladesh—comprising the largest visible minority group in Canada (25.6%) [1]. This number is projected to reach approximately 4.1 million by 2036, of which nearly 1 million will be women of child-bearing age [2]. Evidence collected over the last 20 years suggests that cardiovascular disease (CVD) rates in SAs are higher than other ethnic groups living in Canada [3,4,5,6], and, among SAs, CVD is more severe, presents at younger ages, and in some contexts, this is due to differential access to diagnostic and treatment services compared to non-SAs [7,8,9]. The etiology of this elevated risk for CVD in SAs is not fully understood but early life fetal and childhood exposures have been proposed as major determinants.

The aim of this review is to provide an understanding of the current knowledge around perinatal nutrition and gestational diabetes among the high-risk South Asian population as well as summarize work investigating the role of culturally tailored nutrition advice to South Asian women living in high-income settings like Canada. We begin with describing these qualitative and quantitative studies, both completed and underway, and conclude with a description of the design of a randomized trial of a culturally-tailored personalized nutrition intervention to reduce gestational glycaemia in South Asian women living in Canada and its implications.

## 2. Early Life Determinants of CVD

The “Barker hypothesis” proposed in 1990 by epidemiologist David Barker (1938–2013) posits that in humans, intrauterine growth retardation, low birth weight, and premature birth have a causal relationship with the origins of hypertension, type 2 diabetes, and coronary heart disease in adulthood [10]. One long-term consequence of inadequate early nutrition is the impaired development of the endocrine pancreas, which thereafter makes the infant nutritionally “thrifty”. If the post-natal environment is nutritionally deprived, the phenotype is advantageous, but if the same infant is exposed to a high-nutrition post-natal environment, the result may be increased cardiometabolic risk [11].

Two observations provided the impetus for the development of Barker’s hypothesis. Barker and Osmond reported a positive association between a county’s neonatal death rate (a surrogate for low birth weight (LBW) and its cardiovascular mortality rate [12]. In 1989, Barker revisited Hertfordshire County birth records from 1911 to assess the association between birth weight and ischemic heart disease and found that LBW babies had three times the rate of ischemic heart disease as normal-weight babies [10,13].

Children who experience intrauterine growth retardation experience higher risk of coronary heart disease (CHD), some of whom experience catch-up growth early in life. To examine whether catch-up growth during childhood modifies the increased risk of death from CHD associated with reduced intrauterine growth, Eriksson et al. followed 3641 boys from birth through adulthood to assess the association between birth weight and death from CHD, in the context of catch-up growth [14]. Men who died from CHD had an above average body mass index at all ages from 7 to 15 years. They found that those who were born large but were small at age 11 were at no increased risk of adult CHD, whereas those who were born small and experienced large catch-up growth were at the highest risk of death from CHD later in adulthood. The highest death rates from CHD occurred in boys who were thin at birth, but whose weight caught up with the result of an average or > average body mass from the age of 7 years onwards [14].

The Dutch Hunger Winter occurred in West Netherlands near the end of World War 2 when food rations were limited to fewer than 800 kcal/d [15,16]. In a historical cohort study, 300,000 19-year-old men whose mothers were exposed to the Dutch Hunger Winter pre- and post-natally were examined at military induction to test the hypothesis that prenatal and early postnatal nutrition determines subsequent obesity [17]. The influence of maternal exposure to the famine was highly dependent on the timing. Exposure to famine during the last trimester of pregnancy and the first months of life produced significantly lower obesity rates and better glucose tolerance. This is consistent with the inference that nutritional deprivation affected a critical period of development for adipose-tissue cellularity. Exposure to famine during the first half of pregnancy, however, resulted in significantly higher obesity rates and poorer glucose tolerance. These data are supported by additional famine studies from China, Ukraine, and Austria but not those from the former Soviet Union or those undertaken in Africa [18,19,20,21].

Nutrition is a critical lifestyle-related factor that influences the development of the fetus before birth and infant and child health in early life. In low-income countries, health risks are primarily due to undernutrition, while in high-income countries such as Canada, nutrition-related health risks arise primarily from overnutrition (e.g., excess calories), most strikingly for obesity and associated NCDs [22]. These poor-quality diets can result in the paradoxical “overfed-undernourished” expectant mothers, and such unbalanced nutrition is associated with several adverse maternal, infant, and child outcomes. These include excessive weight gain during pregnancy, gestational diabetes mellitus (GDM), slow or rapid postnatal growth, infant and childhood adiposity, allergic disorders, and asthma. Data on developmental activities during fetal life underscore the critical influence of maternal diet on fetal growth and development. Interactions between genetic and epigenetic factors (in both mother and fetus), and sub-optimal maternal nutrition, may increase infant susceptibility to adverse health outcomes like adiposity, metabolic-syndrome-related factors, allergic disorders, and asthma [23,24,25]. Furthermore, infant feeding patterns and alterations in the gut microbiota during early years may accelerate the development of these adverse health outcomes longitudinally [26,27,28].

## 3. Gestational Diabetes and Related Complications to Mother and Offspring

GDM is a condition in which a woman without diabetes develops high blood glucose levels during pregnancy [29]. There are several proximal and distal complications associated with GDM for both the mother and her offspring. Women who are diagnosed with GDM have a higher risk for perinatal complications including hypertension, pre-eclampsia, polyhydramnios, caesarean section, and shoulder dystocia [30,31]. A French cohort consisting of 796,346 deliveries, including 57,629 women with GDM, showed that the risk for caesarian section was forty percent higher (OR = 1.4; 95% CI: 1.4, 1.4), while the risk for pre-eclampsia was seventy percent higher (OR = 1.7, 95% CI: 1.6, 1.7) in women with GDM compared to those who did not have GDM during pregnancy [32].

Concurrently, the offspring of these women are at a higher risk for macrosomia or increased fetal size, large for gestational age phenotypes, neonatal hypoglycaemia, preterm birth, shoulder dystocia, and respiratory distress [30,31]. According to the data from the French cohort, the odds of such risks are more than double, with the odd for pre-term birth being thirty percent higher (OR = 1.3; 95% CI: 1.3, 1.4) and the odds for macrosomia being eighty percent higher (OR = 1.8; 95% CI: 1.7, 1.8) in offspring of women who were diagnosed with GDM compared to those who did not have GDM during pregnancy [32].

GDM is a strong risk factor for type 2 diabetes (T2DM). Women diagnosed with GDM, compared to those without, have a seven-fold higher lifetime risk of T2DM, and 50% of them develop T2DM within five years of giving birth [33,34,35,36,37,38,39]. GDM predicts future atherosclerosis and cardiovascular disease (CVD) in the mother and increases the risk of T2DM in her offspring up to eight-fold [35,36,37,38,39,40,41,42,43]. SA women have double the risk of GDM of white European women. Their offspring also have increased risk factors for future T2DM, including higher birth weight, more adipose tissue, and reduced insulin sensitivity. These risk factors are appreciably more common in SA infants born to mothers with GDM than infants born to mothers without GDM [44]. We have shown that 36.3% of SA women in Ontario develop GDM [44]. In pregnant SA women, a poor-quality diet during pregnancy increases the chances of developing GDM (odds ratio [OR]: 1.62; 95% CI: 1.20 to 2.19) [44]. Indeed, the finding that ≈13% of GDM cases in this population could be prevented if diet quality was not poor (population attributable risk [PAR:]: 12.8%) [44] highlights the preventive value of an intrapartum dietary intervention to reduce GDM [42,43]. Improving diet quality in the Canadian population of SA pregnant women may therefore prevent up to 6500 new T2DM cases by 2031 [2,44]. The next two sections will highlight current findings about the role of GDM in newborn and childhood health from observational birth cohorts as well as interventional studies of SAs.

## 4. Birth Cohort Studies for Understanding GDM in South Asian Populations

Birth cohort studies collect information on an individual at or before a person’s birth and continue to study the same individuals after birth at regular intervals [45]. In these observational studies, there is no randomization to exposure groups and no attempt to manipulate exposure status. They vary in sample size from large studies that aim to be nationally representative to those that are area-based and may enrol many hundreds to many thousands of participants. Such studies are a great resource when studying early determinants of health. Birth cohorts can be useful in elucidating determinants of health that originate in utero and can differentiate the impact of these from other exposures accumulated throughout one’s lifespan during other critical windows of exposure (e.g., childhood). Birth cohorts are a relatively new form of observational studies that originated in response to the DOHaD hypothesis [46]. Apart from highlighting the risk factors that contribute to perinatal diseases, birth cohorts can also inform interventions developed to address such risk factors and improve the long-term health of mothers and their offspring, especially when accompanied by qualitative studies, as we describe later in the paper. 

Several birth cohorts have been established over the last few years, both in India [47,48,49,50] and elsewhere [51,52]. Notable among these looking at perinatal nutrition are the Pune Maternal Nutrition Study in India, the Born in Bradford (BiB) Cohort [51] in the United Kingdom, and the South Asian Birth Cohort (START) in Canada [52]. Here we summarize the findings from these cohorts related to GDM in SA women.

### 4.1. Pune Maternal Nutrition Study Cohort (PMNS)

The Pune Maternal Nutrition Study (PMNS), one of the largest birth cohorts in India, was initiated in 1994. Much of the data that has highlighted the importance of fetal programming in SAs comes from the PMNS cohort. Designed to assess how maternal dietary practices affect fetal growth and diabetes, the PMNS recruited *n* = 2675 married women of childbearing age from six villages near Pune, India between 1994 and 1996 [50]. In total, 797 women became pregnant and were followed up. Cultural differences between maternal diet and health were identified by comparison with 668 white European mother-infant pairs in Southampton, United Kingdom (UK).

Findings from the PMNS cohort show that SA newborns are, on average, 800 g lighter and smaller in size (abdominal circumference deficit: SD score = −2.99; 95% CI: −3.09 to −2.89) but have similar visceral adiposity (VAT) (subscapular thickness deficit: SD score = −0.53; 95% CI: −0.61 to 0.46) as white European newborns from the UK [50,53]. Such findings have highlighted the prevalence of the “thin-fat” phenotype in SA. Other data from this cohort suggest that several genetic and environmental factors may contribute to this phenotype. One such study, conducted using data from the PMNS, showed that maternal malnutrition, specifically vitamin B12 deficiency, is significantly associated with insulin resistance in their offspring at 6 years of age [54]. Moreover, findings from the PMNS suggest that this insulin resistance in early childhood may be attributable to the increased glucose (8.1 mmol/L vs. 7.5 mmol/L; P = 0.01) and insulin concentrations (321 pmol/L vs. 289 pmol/L; P = 0.04) at birth in children who were born with low birth weight compared to the ones who weighed >3.0 kg [55,56]. These data are consistent with Barker’s hypothesis.

### 4.2. Findings from the South Asian Birth Cohort (START)

The South Asian Birth Cohort (START) study is a prospective birth cohort, established in 2012. It is designed to evaluate the genetic and environmental risk factors that contribute to metabolic disorders in SA women and their offspring [51]. This cohort study recruited 1012 SA pregnant women between the ages of 18 and 40 years old during their second trimester of pregnancy, with a sister cohort (with similar dimensions) developed in India [51] (Appendix A).

Findings from this cohort suggest that the prevalence of GDM in the SA population of Ontario, Canada is 36.3%, with maternal age, height, pre-pregnancy weight, diet quality, and family history of T2DM being identified as risk factors [44]. The cohort has also shown that, like their Pune counterparts, SA newborns in Canada have significantly lower birth weight (3.3 kg vs. 3.5 kg; P = 0.001) and higher skinfold thickness (11.7 mm vs. 10.6 mm; P = 0.0001) compared to white European newborns, and this is associated with higher maternal glucose and adiposity in SA women [57].

### 4.3. Findings from the Born in Bradford (BiB) Cohort

Another large international birth cohort outside the South Asian sub-continent studying in-utero determinants of health, the Born in Bradford (BiB) Cohort was established in 2007. The BiB cohort aims to assess the environmental and genetic determinants of health in SAs residing in Bradford, a northern UK city. The cohort recruited 12,453 women and 13,776 offspring, roughly 50% of whom are of Pakistani origin [52]. These women were recruited from the Bradford Royal Infirmary between 2007–2010 at their 26th to 28th week pregnancy visit.

Findings from the BiB cohort suggest that SA newborns have a lower mean birth weight than white European newborns [58]. Moreover, data from the cohort has shown that this low birth weight in SA newborns increases their risk for several morbidities in later life such as an increased number of antibiotic, analgesic, and bronchodilator prescriptions used, increased incidence of GP consultations, and emergency or elective hospital episodes [58].

## 5. Randomized Trials for GDM Prevention

Non-pharmacological trials to prevent GDM, including multidimensional coaching, [56,57], intensive dietary counselling [59,60], and/or physical activity promotion [61], are few, frequently non-randomized, and have heterogeneous designs and outcomes [62,63]. The current evidence does not suggest an optimal prevention strategy [64,65]. The complexity of healthy behaviour interventions, the variability of adherence and delay before introduction, and the heterogeneity of the maternal metabolic profile and diagnostic criteria in GDM are design factors that contribute to discrepant results.

Diet-based interventions appear to demonstrate the most potential for GDM prevention [66,67,68], as indicated by our network meta-analysis (21 trials during pregnancy; *n* = 1865 women) [69]. We found that reduced glycaemic load and/or increased fiber intake, with appropriate gestational weight-gain advice, contributed to improved fasting glucose levels. In another meta-analysis (44 trials; *n* = 7278 women) [70] diet modification alone (as assessed in three trials; *n* = 409 women) reduced GDM (relative risk [RR]: 0.68; 95% CI: 0.48 to 0.96). Following the publication of our review, there have been two other important studies in the area. One randomized controlled trial in Ireland (*n* = 565) found that exercise and nutrition with smartphone support did not reduce GDM in overweight and obese women randomized at a mean gestational age of 15.3 weeks (RR: 1.1; 95% CI: 0.71 to 1.66) [71]. However, this study aimed to modify only one component of their diet (i.e., a reduced glycaemic index). A second trial (Spain; *n* = 1000) found that a dietary pattern, i.e., a Mediterranean diet supplemented with extra virgin olive oil and pistachios reduced the incidence of GDM (RR: 0.73; 95% CI: 0.56 to 0.95) compared with standard low-fat dietary advice [72]. Both studies enrolled mostly white European women. There has been a trial of diet to prevent GDM in India (OR: 0.56; 95% CI: 0.36 to 0.86), but it is not directly relevant to SAs that have emigrated, as the subjects were mainly women living in slums in Mumbai [73]. These data highlight the importance of rigorously testing tailored dietary interventions in a Canadian SA population.

## 6. Laying the Groundwork for Culturally Tailored Interventions: The Canadian Experience

Building on the START study, our research group has been actively investigating the role of culturally tailored nutrition advice to provide SA women living in high-income countries (HIC), such as Canada. Below, we describe some of these studies, both completed and underway (Figure 1 and summarized in Appendix A).

### 6.1. Qualitative Studies

#### 6.1.1. The START Grandmothers’ Study

Grandmothers play important roles in supporting the family unit, especially in relation to perinatal health and child-rearing. To better understand the health perceptions of Canadian SA grandmothers, we used constructivist grounded theory to sample and interview 17 SA grandmothers whose residences were in Ontario, Canada [74]. Interviews were audio-recorded, transcribed verbatim, and coded/analyzed using a staged approach. We found that many grandmothers believed that:the pre-conception phase should emphasize the establishment of healthy habits (nutrition, physical activity, and mental wellness);the gestational period should encompass an enriched environment (positive relationships, healthy routines, nutritional enhancement) and;the postpartum period should focus on healing and restoration for the mother and newborn child (self-care, bonding, rebuilding healthy habits).

Many of the grandmothers described these three phases as a cyclical relationship where healing and restoration transitions to re-establishing healthy habits before a new pregnancy is initiated. Grandmothers also took initiative in supporting their daughters and/or daughters-in-law and actively encouraged perinatal health communication (Appendix A).

#### 6.1.2. DESI-GDM Qualitative Study

A UK study identified that many SA women experience pregnancy as a stressful period and perceive exercise and diet restrictions to worsen specific symptoms, particularly fatigue [75]. Notably, when confronted with a GDM diagnosis, pregnant SA women were challenged in understanding their diagnosis and implementing responsive lifestyle modifications [76]. They seemed to maintain routine dietary habits and activities, including those identified as traditional [76].

In the “Culturally-tailored personalized nutrition intervention in South Asian women at risk of Gestational Diabetes Mellitus Qualitative Study” (DESI-GDM-Q), we conducted interviews and focus groups with 10 pregnant or recently pregnant SA women to allow women to foster open discussion and identify specific barriers to prescribed dietary advice adherence. The interview schedule included questions designed to elicit beliefs about optimal health behaviours, with a focus on diet and physical activity, for women before and during pregnancy. Interviews and focus groups were also conducted with 11 healthcare providers who self-identified as counselling pregnant SA women in their respective practices. These semi-structured interviews were conducted by female SA interviewers and a female healthcare professional. The interviews were transcribed verbatim and translated into English when necessary. We used an inductive process to analyze interviews and focus group discussions [77]. Data collection and analysis occurred iteratively and concurrently, and group discussions were held to reach an agreed point of saturation (Appendix A).

The themes that emerged from the perspectives of SA pregnant women included: (1) locus of control and (2) information-seeking, while the themes identified among healthcare providers included: (1) cultural competency and (2) clinic management and resources. An overall theme shared by SA pregnant women and their healthcare providers was the importance of the provider being competent in cultural awareness relevant to SA women.

The reflected intersection of the needs, perceptions, and experiences of pregnant women and healthcare providers illuminate opportunities for the development of effective and meaningful interventions. For example, consultations with diverse stakeholder groups can guide the co-creation of a culturally tailored and evidence-based diet and physical activity intervention to achieve normal levels of blood glucose during pregnancy. A culturally tailored approach can provide program planners and researchers deeper insight into end-user preferences for content, format, and preferred delivery method and help understand and overcome misconceptions while facilitating increased program participation. A better understanding of these cultural underpinnings can support the development of successful interventions tailored for pregnant SA women or their healthcare providers.

#### 6.1.3. The HAPPY Study

In young children, poor diet quality and physical inactivity contribute to their risk of excess body weight and metabolic syndrome (MetS). Weight excess, occurring in the first years of life, tends to persist long-term. To better understand early-life family and home environment contributors to this, we designed the study of Health Behaviour-Related Perception, Attitudes, and Practice of Families with Young Children (HAPPY) to better understand the experiences and perspectives of SA mothers in terms of healthy active living lifestyles. This qualitative descriptive study used a thematic analysis approach to explore salient themes arising from the data. In-depth, face-to-face interviews revealed important barriers to and facilitators of healthy active living, as faced by young SA families living in the Peel Region (Appendix A). Barriers to healthy eating included: lack of time for engaging in healthy food preparation, lack of knowledge about healthy eating, viewing healthy eating as a time-limited diet, spouses/children’s unhealthy eating habits, and personal pressures to eat unhealthy foods; facilitators included: setting clear goals, better access to fresh vegetables and fruits, clear arrangements for food preparation, and engaging in vegetarian/vegan ways of eating. Barriers to physical activity included: lack of time and energy to engage in exercise, competing priorities, lack of childcare, limited access to relevant programs; facilitators included: viewing exercise as enjoyable and stress releasing, use of tracking devices, and family support.

### 6.2. Mixed Methods Research Studies

#### 6.2.1. START WATCH

Many SA women can act as agents of change who can reinforce and reiterate healthy active living messages (dietary, sleep, physical activity) within their social networks. We designed the START “Women as Agents of Change” (START WATCH) study to better understand the arrangement of social networks among young SA families living in Peel Region. Our aim is to assess if structural supports (e.g., allied health professionals, programs in temples, schools, workplaces, etc.,) can empower SA women as agents of change. We will conduct a social network analysis (SNA) and exploratory case study to comprehensively explore and describe how SA women in the Peel Region promote healthy active living lifestyles within their families and social networks (Appendix A). We will observe how messages are shared and use quantitative and qualitative data collection to further explore how social networks function and influence members. To supplement this analysis, we will conduct a community readiness model to assess the Peel Region’s “stage of readiness.” This assessment will help predict how willing and prepared a community is to address lifestyle change as it relates to NCD prevention.

#### 6.2.2. SMART START

The SMART START study uses a mixed-methods multiple case study design to evaluate an evidence-based, conceptually informed arts-based prenatal knowledge translation (KT) intervention (a KT tool called ‘SMART START’) (Kandasamy et al., in progress). We built upon the knowledge gaps introduced by the DESI-GDM qualitative study to develop communication points tailored uniquely for patients and healthcare practitioners. The ‘SMART START’ KT tool, a tri-part multi-media toolkit consisting of an informational video narrative, illustrated booklet, and summary material was evaluated against the College of Family Physicians standard care “Prenatal Health Notes.” In partnership with family practices, ‘SMART START’ was evaluated primarily for feasibility, acceptability, and interest and secondarily for change in knowledge, attitudes, practices, and confidence (Appendix A). It was demonstrated that healthy active living is a topic area with a high degree of interest among pregnant people and their family physicians; we showed feasibility regarding good, consistent recruitment (20 patients recruited over a 7-month period) and retention (100% retention; reason for inability to follow-up was due to miscarriage). Overall, pregnant patients valued the new prenatal information they learned, improved some of their physical activity and dietary behaviours, and made recommendations for improving future prenatal health communication.

## 7. Design and Piloting an RCT

Building upon our findings in START [44], which identified diet as a key modifiable risk factor for GDM, DESI-GDM Qualitative, and SMART START [1,78,79], we have developed a dietary intervention program for South Asian pregnant women living in Canada (Appendix A).

Our dietary intervention is focused on:Providing personalized food recommendations that consider a woman’s current dietary habits by identifying food choices and substitutions that will optimize her diet;Providing dietary advice that is sensitive to religious and regional culinary practices;Involving the household meal preparer, if this is not the participant herself, in the coaching contacts;Use of mobile health technology to support self-management and reduce the amount of in-office time a healthcare practitioner spends on dietary counselling.

## 8. The DESI-GDM Randomized Controlled Trial

DESI-GDM is a 2-arm parallel RCT to assess the impact of a culturally tailored, personalized nutrition intervention on glycaemic response to an oral glucose load (as measured by the area-under-the curve glucose) in high-risk pregnant SA women. As of 24 February 2021, no trials in a SA population outside of India are registered with ClinicalTrials.gov or the International Register of Clinical Trials other than our current and previous pilot study [80].

### 8.1. The DESI-GDM RCT Pilot

The pilot study’s purpose was to assess the feasibility of recruitment, randomization, intervention adherence, and participant retention. Success criteria defined for recruitment was >6 participants/month. The clinical outcomes of interest were the glycaemic response to an oral glucose load, incidence of GDM, and infant birth weight. Between 1 March 2019 and 31 July 2019, we recruited participants through family doctors, family health clinics, OB/GYN clinics, and through posters in ethnic grocery stores, community centers, and temples. To be eligible for this trial, women had to be of SA origin in the 2nd trimester of pregnancy and possess two of the following risk factors: age > 29, low diet quality, a family history of DM or prior gestational diabetes, or a pre-pregnancy BMI > 23.

We screened 34 participants, and 20 women of SA ancestry with risk factors for gestational diabetes (GDM) were enrolled over 133 days, resulting in a recruitment rate of 4.4 participants per month. The average age of participants was 30.6 ± 4.0 years, at 14.8 ± 2.1 weeks’ gestation, and they had lived in Canada for 8.6 ± 7.8 years. Half of the women had a family history of diabetes; 15% had previous gestational diabetes, and 85% had a BMI > 23. Their average diet quality score was 0.8 ± 0.9 out of 6, using a metric that was associated with GDM in our SA birth cohort study [44]. Scores that approached 0 indicated low diet quality, and scores that approached 6 indicated high diet quality.

Each participant assigned to the intervention group met with a health coach via phone or in-person to set and support the attainment of nutrition goals, 2–4 times, within the context of energy-balance for recommended gestational weight gain and personal values and preferences. Between 2 and 4 “SMART” goals were set according to the following principles: (1) replacing traditional fried foods and meat with more vegetable protein and raw and cooked vegetables; (2) reducing carbohydrate intake, for those who report consumption of high quantities of refined carbohydrates; (3) improving carbohydrate quality (replacing high glycaemic index, refined-grain SA foods with lower glycaemic index, whole-grain SA foods (e.g., replacing white rice with parboiled rice, brown rice, or legumes; replacing white flour paratha with whole wheat paratha); (4) reducing trans fats (e.g., switching from commercial ghee to butter or vegetable oil); (5) scheduling regular mealtimes, 3–4 h apart; (6) replacing fried, high-carbohydrate snacks (e.g., chaat or dahi vada) with higher protein and fat snacks (such as nuts or full-fat dairy); (7) replacing sweets and sugar-sweetened drinks (sweetened chai, soda pop, and juice) with water or unsweetened tea and choosing fruits and other lower-sugar desserts over desserts high in starch and sugar.

The control group participants were provided with paper copies and website links to “The Sensible Guide to a Healthy Pregnancy”, which provides advice on healthy eating, physical activity, and other lifestyle factors during pregnancy, plus additional materials adapted specifically for the SA community.

All participants completed the OGTT visit, but only 12 (60%) completed the 75-g OGTT as per study protocol. The remaining eight participants completed their general-practitioner-prescribed the 50-g test.

The pilot study led us to make the following protocol changes for the main trial. First, we found it difficult to identify and enrol participants who were <14 weeks pregnant. Only three of our participants were <13 weeks. Thus, we increased our enrolment window, allowing participants into the study up to 18 weeks’ gestation. Second, of the seven participants who were due for their OGTT, three refused the 75-g OGTT and preferred to follow their family doctor’s order for the 50-g glucose challenge. To avoid this problem in the full trial, family physicians were more closely engaged in the referral and enrolment process, based on an approach that we have used successfully in the START study. Referrals were coordinated with the participant’s care provider such that the requisition for the 75-g OGTT and blood study replaced the test ordered by the provider. Fourth, our participants responded well to the health coach, who was a multilingual SA woman, who could incorporate culturally sensitive dietary information. Though all participants were fluent in English, some words or concepts were better expressed in a SA language, and we will work to match this characteristic in the full trial.

### 8.2. The DESI-GDM RCT

In the DESI-GDM trial, pregnant SA women without known diabetes will be enrolled early in the second trimester. This avoids including women experiencing peak nausea early in pregnancy, and it is past the highest period of risk for spontaneous abortion. Eligible participants will be pregnant women of SA ancestry at gestational weeks 14–18 with a singleton pregnancy; and ≥2 of the following GDM risk factors: age > 29, low diet quality (assessed with a short diet screener [44]), a family history of DM, GDM during a previous pregnancy, or a pre-pregnancy BMI ≥ 23 [44]. We will exclude women with pre-existing type 1 or type 2 diabetes; high blood pressure (≥140 mmHg systolic or ≥90 mmHg diastolic); a poor understanding of English (93% of Canadian SA women in the National Household Survey could conduct a conversation in one or both official languages [1]); those unwilling to modify their diet, based on a screening question; those at high risk of adverse pregnancy outcomes other than GDM (e.g., twins or higher-order multiples, use of fertility treatment, preexisting hypertension, history of placenta previa, or pre-term delivery); or who are enrolled in another study (Appendix A).

A personalized nutrition plan will be developed for each woman by a culturally congruent dietitian. This plan will respect faith-based food choices and regional preferences. The plan will be delivered by a culturally congruent health coach and consider baseline dietary intake, energy-balance for recommended gestational weight gain, personal values, and preferences, through setting 2–4 “SMART” goals (as described above). Our nutrition and behaviour change experts have developed text messages that support 11 categories of nutrition goals, targeted to address eating behaviours identified by participants in our qualitative study, designed to optimize energy balance for weight gain and improve dietary quality. Control group participants will be provided with paper copies of and website links to “The Sensible Guide to a Healthy Pregnancy”, which provides advice on healthy eating, physical activity, and other lifestyle factors during pregnancy, plus additional materials adapted specifically for the SA community. Healthcare providers in the Peel Region use these tools routinely (Diabetes Canada: https://bit.ly/2m8r2tT, accessed on 20 August 2021, or Heart & Stroke: https://bit.ly/2lDubl7, accessed on 20 August 2021). Participants in both control and intervention groups receive at least 1 text message every week. We developed a roster of text messages to encourage walking and healthy eating, which were well accepted by participants in our pilot study.

Text messages to reinforce individual nutrition goals for the intervention group will be sent weekly. We will encourage participants to respond and engage in two-way dialogue with the health coach. This strategy has better uptake than impersonal one-way messages at predetermined times that do not permit confirmation of receipt or feedback [81,82]. The intervention group will receive at least two text messages/week at the times of day requested by the participant: one or more tailored diet text messages and a message that provides one of six walking tips. Coaching calls to the intervention group will be made post-baseline weeks 2, 4, 6, 8, 10, 12, 14, and 16 (bi-weekly up to OGTT). At each scheduled contact, intervention participants will review the agreed-upon diet goals, and assess, on a Likert scale, how often they were able to meet them (ranging from “never” through “all of the time”); the coach will work with the participant to overcome barriers using our brief action planning guide. In our pilot study, health-coaching calls lasted 15–20 min. The individualized nutrition plan will be modified during follow-up by the health coach, according to feedback from the participant, and through a review of the dietary information collected.

The primary clinical outcome of this trial is the glucose area under the curve (glucose AUC). As a measure of glycaemic response, glucose AUC is a continuous measure of the response to a 75-g OGTT that accounts for variations in fasting plasma glucose levels between individuals. It is calculated by the trapezoidal method using the fasting, 1-h, and 2-h glucose [83]. The AUC is superior to a single measure, which, although convenient for diagnosis and treatment, may not provide complete information regarding the processing of plasma glucose after a load [84]. The secondary outcome is GDM, classified using the cut-offs derived in the BiB cohort: fasting glucose ≥5.2 mmol/L or 2-h post-load ≥7.2 mmol/L [85]. These cut-offs were associated with infant birth weight >90th percentile for gestational age and adiposity (sum of skinfold measurements [SSF] >90th percentile for gestational age) in this cohort [85]. Current clinical cutoffs for the 75-g OGTT used to diagnose GDM in the general population are: fasting glucose ≥5.3 mmol/L, 1-h ≥10.6 mmol/L, or 2-h ≥9.0 mmol/L [86]. In Peel, pregnant women usually undergo a 50-g glucose challenge at 24–28 weeks, with a 1-h value ≥7.8 mmol/L being an indication for a 75-g OGTT [87].

We have selected the 75-g OGTT because:it was used to establish SA-specific diagnostic criteria for GDM, and thus our outcomes will be directly comparable [85];it avoids the high false negative rate of the 50-g GCT among SAs [51,88];one-step screening has the potential for long-term cost savings [89,90,91];Diabetes Canada recognizes that the one-step strategy can identify a subset of women who would not otherwise be identified as having GDM and who may benefit with regards to certain perinatal outcomes [86].

We will co-ordinate with the healthcare provider to ensure participants receive the 75-g OGTT between 24–28 weeks, avoiding the two-step screen. A study endocrinologist will review the OGTT results and send a letter to the participant’s provider communicating the result, to ensure continuity of care and appropriate management. We will assess the sensitivity and specificity of the BiB definition against the IASPSG or WHO criteria. To evaluate safety outcomes, we measure maternal blood pressure at baseline and the OGTT visit and note any pregnancy complications at coaching contacts.

With 95 participants per group, and an expected 10% loss-to-follow-up, we will have ≈86 participants per group, which will provide 90% power to detect at least a 15% between-group difference in AUC glucose, assuming 70% adherence to the intervention (i.e., goal setting, health coach contacts, tracking) and SD of AUC glucose = 173 mmol/min [44]. A change of at least this magnitude has been observed in trials of fibre supplements [89], high-protein [92], and high-fat diets [93]. We will have 80% power to detect a 61% relative risk reduction in GDM. An effect of this magnitude is unlikely, so we will look to see if GDM is lower in the intervention. Failure to observe substantial impact on AUC glucose (i.e., <10% reduction) with no signal of benefit for GDM (i.e., an OR between 0.9 and 1.1) or a signal of harm (OR > 1.1) with high intervention fidelity will suggest that a larger trial primarily powered to detect differences in GDM is not warranted.

## 9. Conclusions

In summary, maternal lifestyle factors influence the in utero environment of developing fetuses, which in turn influences the health outcome of newborns. A considerable amount of data exists demonstrating the early life risk factors for CVD among SAs. Trials of diet and behaviour modification to reduce gestational glycaemia in pregnant SA women in Canada are needed because:SAs are among the fasting growing and largest non-white minority group in Canada [1,2];SAs are a high risk population for GDM, T2DM, and CVD [94];GDM is a risk factor for future T2DM and CVD in both mother and offspring (producing two high-risk individuals) [36,37,38,39,40,41,42];There have been no prior or planned RCTs of GDM prevention in SAs living in high-income countries as of 1 May 2021 other than our pilot study [80];Pregnancy is an ideal time to intervene with a diet intervention [95].

We have designed an RCT that addresses key aspects shown to improve the effectiveness of lifestyle interventions during pregnancy: we target a high-risk population, initiate the intervention early, provide the correct intensity and frequency of contact, and manage gestational weight gain [96]. Groundwork for the trial’s success includes our cohort study identifying diet as a modifiable risk factor for GDM in SAs [44], a qualitative study to understand the unmet needs of the population [75], pilot data showing that participant recruitment and retention is feasible and that the delivery of the intervention is acceptable, and strong partnerships with healthcare providers and public health in Peel, a region with a high number of SAs. Our trial will assess whether diet intervention reduces gestational glycaemia. The intervention targets two at-risk individuals: mother and infant, “breaking the cycle” of maternal gestational dysglycaemia, excess infant adiposity and insulin resistance, and CVD in both mother and baby.

## Figures and Tables

**Figure 1 nutrients-13-02932-f001:**
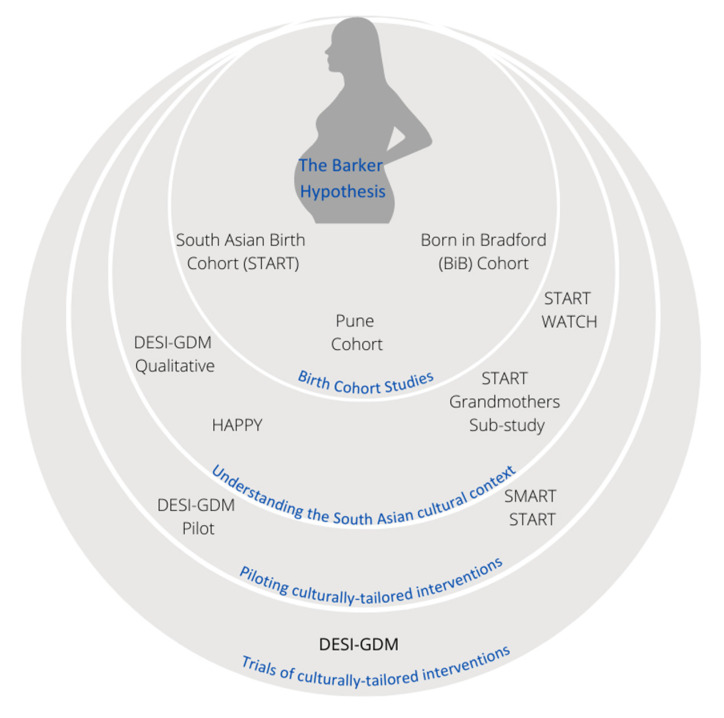
The evolution of SA perinatal health in South Asian women research, inspired by the Barker hypothesis. The Barker hypothesis gave rise to birth cohort studies, which encouraged the development of culturally tailored interventions and evaluations.

## Data Availability

This article reviews previously published work, as well as describes ongoing studies. The data generated and/or analyzed as part of these ongoing studies are not publicly available because participants in the described studies did not consent to public sharing of their data at the time of recruitment. Datasets may be made available from the corresponding author on reasonable request.

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
