# Peer review of "Studies to Improve Perinatal Health through Diet and Lifestyle among South Asian Women Living in Canada: A Brief History and Future Research Directions"

_nutrients, 2021, doi:10.3390/nu13092932_

Round 1

Reviewer 1 Report

In this paper, Desai et al describe qualitative and quantitative studies, both completed and underway. They conclude with a description of the design of a randomized trial of a culturally tailored personalized nutrition intervention to reduce gestational glycemia in South Asian women living in Canada, and its implications.
The review is detailed and well up-to-date, but lacks the necessary interest that would entail extending it to broader and more heterogeneous populations, representative of current societies in developed countries.
The manuscript does not propose innovative or impactful actions in the prevention of gestational diabetes mellitus and the announcement of a new trial on the key aspects to improve the effectiveness of lifestyle on gestational blood glucose.
The work would improve remarkably if it were carried out as a systematic review and meta-analysis of the published works on the impact of lifestyle on the prevention of gestational diabetes mellitus.

Author Response

Dear Reviewer 1,

Thank you for your review and feedback. Please find our responses below:

  • In this paper, Desai et al describe qualitative and quantitative studies, both completed and underway. They conclude with a description of the design of a randomized trial of a culturally tailored personalized nutrition intervention to reduce gestational glycemia in South Asian women living in Canada, and its implications.

  • The review is detailed and well up-to-date, but lacks the necessary interest that would entail extending it to broader and more heterogeneous populations, representative of current societies in developed countries.

Response: Thank you for your feedback. The current review is designed to give an overview of the early life determinants of cardiovascular disease, specifically the role of perinatal nutrition and gestational diabetes among a high-risk population, South Asian women. Our focus on the South Asian population is due to the high prevalence of GDM among South Asian women as well as the higher risk of developing cardiovascular diseases and metabolic risk factors among South Asian people at an earlier age compared to non-South Asians. We feel that the learnings from this review is applicable to other high risk populations.

  • The manuscript does not propose innovative or impactful actions in the prevention of gestational diabetes mellitus and the announcement of a new trial on the key aspects to improve the effectiveness of lifestyle on gestational blood glucose.

Response: Thank you for your feedback. In this review we demonstrate the current knowledge around gestational diabetes, it’s complications to the mother and fetus as well as previous randomized clinical trials of GDM prevention. In doing so we demonstrate that there are no RCTs for GDM prevention among the high risk South Asian women. We then describe the work of our group beginning with the quantitative observation birth cohort (START), leading to qualitative (Grandmother’s Study, DESI-GDM Qualitative study, HAPPY) and mixed methods research (SMART START, START WATCH) studies that ultimately informed the design of DESI-GDM RCT.

  • The work would improve remarkably if it were carried out as a systematic review and meta-analysis of the published works on the impact of lifestyle on the prevention of gestational diabetes mellitus

Response: We appreciate your feedback and the suggestion for a systematic review with meta-analysis of the published work. However, this would have to be the subject of a new manuscript as designing and conducting a robust systematic review and meta-analysis would require a few months of work.

Reviewer 2 Report

Thank you very much for giving me the opportunity to review the manuscript entitled “Studies to improve perinatal health in South Asian women living in Canada: a brief history and future research directions”

Please find my specific comments below:

  • I would recommend to change the title of this manuscript – to indicate that this study focus on dietary issues and make it more suitable to the journal
  • The abstract needs to be rewritten to indicate clearly the aim of this study and the results and conclusions from the review/search and for future directions – in the current for it is not informative at all
  • In the sentence “This number is projected to reach ~ 4.1 million by 2031….” – should it be 2031 or 2036?
  • I would change “Nutrition is a critical environmental factor…” into “Nutrition is a critical lifestyle-related factor”
  • The aim of the study needs to be clearly stated – in the current form it is really difficult to find out what is the rationale/purpose of this manuscript

Author Response

Dear Reviewer 2,

Thank you for your review and feedback. Please find our responses below:

  • I would recommend to change the title of this manuscript – to indicate that this study focus on dietary issues and make it more suitable to the journal

Response: We have considered your feedback and revised the title of our study as per below:

“Studies to improve perinatal health through diet and lifestyle among South Asian women living in Canada: a brief history and future research directions”

  • The abstract needs to be rewritten to indicate clearly the aim of this study and the results and conclusions from the review/search and for future directions – in the current for it is not informative at all

Response: Thank you for your feedback. We have revised the abstract to clearly state the aim of our paper, as per below:

The aim of this review is to provide an understanding of the current knowledge around perinatal nutrition and gestational diabetes among the high-risk South Asian population as well as summarize our research activities investigating the role of culturally-tailored nutrition advice to South Asian women living in high-income settings such as Canada.

  • In the sentence “This number is projected to reach ~ 4.1 million by 2031….” – should it be 2031 or 2036?

Response: Thank you for pointing to this to us. We have corrected to the year to 2036.

  • I would change “Nutrition is a critical environmental factor…” into “Nutrition is a critical lifestyle-related factor”

Response: We agree with your suggestion and have made the change.

  • The aim of the study needs to be clearly stated – in the current form it is really difficult to find out what is the rationale/purpose of this manuscript

Response: The aim of the paper has been stated in in the introduction now.

“The aim of this review is to provide an understanding of the current knowledge around perinatal nutrition and gestational diabetes among the high-risk South Asian population as well as summarize our work investigating the role of culturally tailored nutrition advice to provide South Asian women living in high-income settings like Canada. We begin with describing these qualitative and quantitative studies, both completed and underway, and conclude with a description of the design of a randomized trial of a culturally tailored personalized nutrition intervention to reduce gestational glycemia in South Asian women living in Canada, and its implications.”

Sincerely,

Russell de Souza

Round 2

Reviewer 1 Report

I believe that the manuscript has improved remarkably with the changes made by the authors and that it could be considered for publication as a review in Nutrients.

Reviewer 2 Report

The authors have corrected the manuscript according to my comments.